# A Deep Learning Model with a Self-Attention Mechanism for Leg Joint Angle Estimation across Varied Locomotion Modes

**DOI:** 10.3390/s24010211

**Published:** 2023-12-29

**Authors:** Guanlin Ding, Ioannis Georgilas, Andrew Plummer

**Affiliations:** Department of Mechanical Engineering, University of Bath, Bath BA2 7AY, UK; gd442@bath.ac.uk (G.D.); a.r.plummer@bath.ac.uk (A.P.)

**Keywords:** trajectory planning, transformer, attention mechanism, prostheses, transfer learning

## Abstract

Conventional trajectory planning for lower limb assistive devices usually relies on a finite-state strategy, which pre-defines fixed trajectory types for specific gait events and activities. The advancement of deep learning enables walking assistive devices to better adapt to varied terrains for diverse users by learning movement patterns from gait data. Using a self-attention mechanism, a temporal deep learning model is developed in this study to continuously generate lower limb joint angle trajectories for an ankle and knee across various activities. Additional analyses, including using Fast Fourier Transform and paired *t*-tests, are conducted to demonstrate the benefits of the proposed attention model architecture over the existing methods. Transfer learning has also been performed to prove the importance of data diversity. Under a 10-fold leave-one-out testing scheme, the observed attention model errors are 11.50% (±2.37%) and 9.31% (±1.56%) NRMSE for ankle and knee angle estimation, respectively, which are small in comparison to other studies. Statistical analysis using the paired *t*-test reveals that the proposed attention model appears superior to the baseline model in terms of reduced prediction error. The attention model also produces smoother outputs, which is crucial for safety and comfort. Transfer learning has been shown to effectively reduce model errors and noise, showing the importance of including diverse datasets. The suggested joint angle trajectory generator has the potential to seamlessly switch between different locomotion tasks, thereby mitigating the problem of detecting activity transitions encountered by the traditional finite-state strategy. This data-driven trajectory generation method can also reduce the burden on personalization, as traditional devices rely on prosthetists to experimentally tune many parameters for individuals with diverse gait patterns.

## 1. Introduction

The ability to walk can be diminished by amputation, aging, injury, or neurological and musculoskeletal diseases, such as osteoarthritis [1,2]. Around 4% of the global population experiences difficulties with mobility in their daily life [3]. To tackle the need to improve mobility, it is crucial to enhance the intelligence and performance of assistive devices, including active prostheses, orthoses, and exoskeletal systems. In accordance with the generic control architecture described in [4], powered walking assistive devices require three-level controllers, consisting of an activity detection system, a trajectory generator, and a lower-level execution layer. Within the realm of position-controlled lower limb assistive devices, the core challenge, joint trajectory planning, is receiving increased research effort to develop a continuous and natural walking pattern. The estimation of lower limb joint angles has been studied by many groups [5,6,7] to achieve better trajectory planning for powered assistive devices. Reliable human kinematics prediction is of significance, not only for regaining a healthy walking pattern, but also for facilitating accurate clinical diagnosis and analysis. The assessment of limb joint angles during activities in daily living can reflect the severity of degenerative diseases and neuromuscular disorders [8]. Moreover, human motion estimation is becoming more valued in other domains, such as sports and manual work monitoring to avoid injuries [9].

The trajectory planning strategies commonly employed for prostheses and exoskeletons involve the utilization of activity classification and a phase-based finite-state strategy. This approach entails the selection of a pre-defined trajectory based on the current activity (e.g., inclined ground) and gait event (e.g., toe-off) [1,10]. Yu et al. [11] (2016) introduced a walking aid mode on flat ground to facilitate the toe push-off phase and subsequent foot dorsiflexion. This mode can be activated by a triggering mechanism that uses a foot strain gauge signal to detect the displacement of the body’s center of gravity. However, this finite-state machine (FSM) approach necessitates the categorization of users’ walking patterns into limited activity modes, as well as the division of the gait cycle into discrete gait phases. Consequently, this approach results in discontinuous motions during the transition stage between the locomotion tasks or gait phases [1]. In addition, the safety of device wearers can also be comprised by the latency and error in the classification of the activity or gait phase [12]. Irrespective of the above transition issues, these fixed pre-determined trajectories adapt poorly to real terrain variations that deviate from the structured walking environment found in laboratory settings [1,13].

Moreover, finite-state-based control strategies inevitably introduce many device parameters, thresholds, and switching rules to refine the division of the gait phases [12,14]. Au et al. [15] conducted experiments to establish device settings and mode transition rules for different gait phase stages. These settings included the determination of the net torque needed at the ankle joint, the pre-defined foot position, and the duration for passive modes. Simon et al. [16] also divided the gait phase of each activity mode into finite states, with each state corresponding to a group of impedance parameters, for a total of 140 tunable parameters. Due to inter-subject variations, the device tuning should be tailored to accommodate individuals of different sizes and gait styles. Therefore, the experimental workload is considerable when it comes to optimizing the parameter space. Even for the same device wearer, the well-tuned fixed trajectory might be problematic for comfort due to the intra-subject variations brought on by fatigue, and health issues [17].

Complementary limb motion estimation (CLME) utilizes sensors mounted on either healthy contralateral limbs or ipsilateral residual limbs to statistically infer the optimal trajectory for the affected leg. This limb-coupling concept is more practical for lower limb prostheses as there is a stronger inter-limb coordination relationship in the joints of the lower limb than the upper limb [10]. This approach reduces the need to set up many device parameters because there is less of a requirement for explicit gait phase division. In contrast to a pre-defined trajectory method, CLME enables a more harmonious human–device interaction [1,2]. Vallery et al. [18] undertook a pilot investigation aimed at validating the efficacy of this bio-inspired method for level walking and staircases. Nevertheless, they only examined the relatively straightforward linear mapping between the amputated and healthy body parts and specific linear predictors for different activities. A unified and non-linear model for lower limb trajectory generation is required to seamlessly deal with numerous activities.

The advancement of deep learning enables a more generalized CLME-based trajectory planner to be created by learning non-linear motion patterns from daily motion data. Interest has been directed towards recurrent neural networks (RNN) [1,19], convolutional neural networks (CNN) [12,20], and dense neural networks (DNN) [21], as the main models for joint angle prediction. Instead of focusing on individual models for each activity, most studies aim to develop a comprehensive deep learning model that can deal with various activity modes. Thus, this continuous estimation approach does not necessitate switching device locomotive mode for different activities explicitly. Moreover, in contrast to a pre-defined trajectory, an appropriate gait pattern can be learnt for each user to adapt to their own stride length and cadence, which alleviates the issues related to inter-subject, intra-subject, and intra-gait variation [10,22,23]. Most of those approaches rely on inertial measurement unit (IMU) data to train the deep learning methods [10,12,20,22,23], while others aim to exploit the use of electromyography (EMG) [10,19,20,21]. The latter indicates that EMGs are more complex to use and better suited for classification tasks.

Among basic deep learning models, the long short-term memory (LSTM) neural network has received widespread interest for estimating joint angles because human locomotion is inherently a time-sequential task [6,24,25]. Although the gate-controlled mechanism of the LSTM network has been found to be advantageous in capturing long-term temporal dependencies [20] (Wang, C. et al., 2022), there still remains a difficulty in extracting more information from long historical contexts [5]. Because of this, researchers have focused their efforts on the attention mechanism to handle longer movement time histories. In their study, Rai et al. [5] adopted a dual-stage attention model to extract temporal features, with two LSTM layers acting as the encoder and decoder. In addition, Ding et al. [26] proposed a more concise attention mechanism with a small multilayer perceptron to weight each timestep within the time series. With the prevalence of the Transformer model [27], the self-attention mechanism is increasingly being applied to a variety of time-series tasks, and some pilot studies have been conducted for human motion estimation as follows. Geissinger and Asbeck [9] utilized a Transformer model with the same implementation as the original design proposed by Vaswani et al. [27] for full-body motion inference, although their model’s performance did not provide the expected advantages. Yin et al. [13] introduced a multi-head self-attention model for gait trajectory prediction, whereby CNN layers were incorporated in this model to further extract the features. However, they predicted the future left thigh flexion angle based on the previous left thigh flexion angles, which is not a useful approach for amputees.

Nevertheless, it remains an open question whether the Transformer architecture, specially designed for natural language tasks, is still as effective for other time-series analysis tasks (e.g., motion estimation) [9,28]. The investigation performed by Geissinger and Asbeck [9] also shows that the original Transformer should be tailored for motion inference tasks. Furthermore, due to its massive network size, the original Transformer model has a long training time and necessitates an extensive database. This study aims to examine the efficacy of a customized self-attention-based model inspired by the original Transformer for continuous lower limb joint angle estimation across various activities. If successful, the model could act as a trajectory generator to provide limb joint angle reference signals for a walking assistive device. The proposed approach will be a step improvement in the existing CLME literature. As an extra step to validate the proposed approach, in addition to comparing model errors between the suggested version and other works, a more robust statistical test is carried out to assess the usefulness of the suggested attention model. Moreover, there are very few studies on the use of transfer learning techniques to improve the performance of joint angle prediction models, a topic that will be investigated in this work. With the learned human motion pattern in the pre-trained model, transfer learning only needs a few new samples to improve the model accuracy. Furthermore, the smoothness and dynamic behavior of model predictions are crucial for safety and comfort. Fast Fourier Transformation (FFT) can assist in analyzing the power distribution of motion signals in the frequency domain, ensuring that critical dynamic behavior is maintained and new frequency components, probably noise, are not introduced. This analysis was conducted to check that the frequency profile of the model predictions was similar to the baseline date.

## 2. Materials and Methods

### 2.1. Data Preparation

The increasing need for gait analysis has led to the creation of many motion capture datasets. These datasets employ a variety of data acquisition techniques, including optical motion capture systems, digital cameras, and wearable sensors [9]. Wearable sensors are more suitable for motion tracking outside the laboratory than optical marker sensing or vision-based techniques [2]. Among wearable methods, EMG-based volitional control is a potential approach for the control of lower limb walking assistive devices, utilizing electrodes to measure muscle activity. However, the use of bioelectric signals, such as surface EMG, requires the careful attachment of electrodes to the participant’s skin surface in a secure manner [21]. It is also important to note that EMG signals exhibit non-stationarity and are affected significantly by noise [29,30]. IMU sensors are comparatively more promising and economical wearable sensors for motion capture tasks. They offer the advantages of convenience and compactness in device placement [2,29]. Therefore, biomechanical kinematics measured by inertial sensors are the main input data used in this study.

Hu et al. [31] released a locomotion dataset that contains IMU-acquired kinematic data, EMG signals, and directly measured joint angles for bilateral lower limb movement. Four 6-axis IMU sensors (MPU-99250; InvenSense, San Jose, CA, USA) were attached to both legs at the thigh and shank, with an additional IMU sensor strapped to the waist. Each IMU sensor consisted of a tri-axial accelerometer and a tri-axial gyroscope. Four electro-goniometers (SG1150; Biometrics Ltd., Newport, UK) recorded the bilateral joint angles from the knee and ankle flexion and the extension in the sagittal anatomical reference frame. EMG signals were also collected in the dataset, but they are not employed in this study due to the mentioned drawbacks. Although the additional input features may be useful to promote model performance, acquiring the signals is challenging for practical walking assistive devices. As for the gait experimental protocol, a group of ten non-disabled participants (ages: 23–29, heights: 160–193 cm, weights: 61–95 kg) were instructed to repeat a pre-determined sequence of movements, several times. The locomotor activities included sitting (S), standing (St), level walking (LW), ramp ascending and descending (RA/RD), as well as stair ascending and descending (SA/SD). Subjects were permitted to walk at a self-decided speed, and their activity transitions were also considered in the activity routine. More details about the data collection and processing can be found in the work by Hu et al. [31].

A few trajectory planners [6,12,19] estimate joint angles for healthy participants utilizing the biomedical signals from the same body side, though this might not be applicable for amputees with an affected lower limb. To avoid such an issue, this work aims to generate the trajectory for the impaired leg by measuring the movement of the contralateral healthy limb. As plotted in Figure 1, fused biomechanical signals, including IMU and goniometer signals, are used from the dataset [31]. Three 6-axis IMU sensors are attached to certain body parts, namely the waist, right thigh, and right shank. These IMU sensors are used to measure the acceleration and angular velocity along three axes, and the resulting IMU kinematic data are taken as the main model input features. The right knee and right ankle angles measured by electro-goniometers in the sagittal plane are also included as model inputs. The corresponding left knee and left ankle angles are targeted as model outputs. The dataset [31] used an original sampling frequency of 500 Hz for both the IMU signals and joint angles, and this study downsampled the data to 50 Hz to save model training time. Downsampling is a frequently employed technique for postprocessing human motion datasets, and 50 Hz is one of the commonly used sampling rates [9].

Before inputting the data into the time-series model, a sliding window spanning 25 timesteps (0.5 s of data) moves through the collected data samples to create time-slice data instances. The resampling process using the sliding window is illustrated in Figure 2, in which one input data instance $x=\left(x^{\langle 1\rangle },x^{\langle 2\rangle },\cdots ,{x^{\langle t\rangle },\cdots ,x}^{\langle T\rangle }\right)$ records multiple timesteps $x^{\langle t\rangle }$, and each timestep $x^{\langle t\rangle }=(x_{1}^{\langle t\rangle },x_{2}^{\langle t\rangle },\cdots ,x_{i}^{\langle t\rangle },\cdots ,x_{m}^{\langle t\rangle })$ contains multiple input features $x_{i}^{\langle t\rangle }$. The generated data instance will be used to forecast the future left ankle and knee angles for the next timestep. Prior to being fed into the model, the input data are normalized, which aids in the model training by eliminating the effect of different magnitudes across the features [2]. The normalization scaling factors (i.e., mean and standard deviation) obtained from the training dataset should be applied to the cross-validation and testing datasets as well [26].

As for the dataset splitting, a 10-fold leave-one-out inter-subject testing scheme is implemented in order to maximally verify the model’s generalization across all subjects. In this scheme, each subject takes turns acting as the testing dataset for ten repetitions; meanwhile, the nine remaining subjects are used to train and cross-validate a new model. This leave-one-out scheme allows the model to evaluate its performance on the unseen subject, hence providing a more accurate representation of the model’s generalization and robustness. Consequently, a total of ten well-trained models were obtained under this 10-fold scheme, and all ten participants have been used as the testing dataset once to validate the effectiveness of the model architecture. This testing strategy avoids choosing a model that provides optimal test performance for only one subject, ensuring a more accurate reflection of the model’s generalizability. Moreover, this cross-subject testing scheme is stricter than the intra-subject testing that uses the training and testing data from the same subject [1]. Consequently, achieving a low model error becomes more challenging when subject diversity is relatively low.

### 2.2. Baseline Model

A gait cycle can be viewed as a periodical temporal sequence, allowing a recurrent neural network to extract long-term dependencies from the gait data [1]. As an enhanced form of RNN, the LSTM neural network is a popular time-series predictor among many biomechanics researchers [2,6,25,32]. Transitional RNN prioritizes the current and prior neighboring timesteps over previous distant timesteps. Due to gradient exploding/vanishing, this shortsighted approach fails to handle a relatively long time series [2]. To overcome this, LSTM introduces a memory cell to deliver information across many timesteps. A gate mechanism is used to assist the memory cell in retaining crucial temporal information, while discarding useless information [26]. When processing the input $x^{\langle t\rangle }$ in the timestep t, as shown in Figure 3, the memory cell will first forget irrelevant information from $c^{\langle t-1\rangle }$ via the forget gate and then be updated by new inputs $x^{\langle t\rangle }$ and the last hidden state $h^{\langle t-1\rangle }$ through the update gate. Then, the new cell state $c^{\langle t\rangle }$ will be delivered to the next timestep. As a result, even the information from very early timesteps can be conveyed to the present moment. Thus, LSTM networks have become widely used in biomechanics estimation, and will be adopted as the baseline model in this study to estimate ankle and knee angles. The corresponding model architecture is depicted in Figure 4.

### 2.3. Attention Model

In order to enhance model performance by better identifying critical inputs, several attention models [5,26] have been used in the estimation of the biomechanical parameters. Recently, the self-attention mechanism employed in the Transformer model [27] has been used for motion inference. An investigation into using the self-attention mechanism to infer the gait pattern will be presented in this paper. The original Transformer architecture was tailored to our task. Several primary components, such as the multi-head self-attention mechanism, positional encoding, residual connection, and feed-forward module, were retained in the proposed customized model, as shown in Figure 5. As suggested by Lim et al. [33], one LSTM layer is used as the first layer to finely extract the temporal correlations in the time-series data from low-level input features.

Assuming there are several timesteps in one input data instance, the self-attention mechanism allows an input timestep to query its importance with other input timesteps, and then the obtained attention scores can be used to weight these input timesteps. The calculation process within the self-attention module is depicted in Figure 6a, and the outputs are computed as:(1)$$\mathrm{Attention}\left(Q,K,V\right)=\mathrm{softmax}\left(\frac{QK^{T}}{\sqrt{d_{k}}}\right)V$$
where Q is the query matrix, K is the key matrix, V is the value matrix, and $d_{k}$ is the dimension of the key matrix.

The matrices Q, K, and V are computed using the same inputs, but different weights as follows:(2)$$Q=AW^{q}$$
(3)$$K=AW^{k}$$
(4)$$V={\mathrm{AW}}^{v}$$
where $W^{q}\;\in \;R^{d_{a}\times d_{k}}$ is the query weight matrix, $W^{k}\;\in R^{d_{a}\times d_{k}}$ is the key weight matrix, and $W^{v}\;\in R^{d_{a}\times d_{v}}$ is the value weight matrix. $A\in \;R^{N\times d_{a}}$ is the input for the self-attention layer, which is also the output of the previous neural network layer in our model.

The multi-head self-attention layer used in the proposed attention model is a modified version of the self-attention mechanism. In this variation, each attention head independently performs the same calculation process as the self-attention mechanism in Figure 6a, but with different weight matrixes, $W_{i}^{q}$, $W_{i}^{k}$, $W_{i}^{v}$, acting as multiple channels to learn different data patterns. The concept of the multi-head mechanism is analogous to the notion of distinct channels in a convolutional neural network. With multiple attention heads, the model’s adaptability to complex input patterns can be improved by capturing various relationships present in the data. Additionally, a multi-head mechanism allows each head to compute in parallel at the same time, hence improving computational efficiency [27]. As shown in Figure 6b, the outcomes of the multiple heads are combined in a concatenated manner as:(5)$$\mathrm{MultiHead}\left(Q,K,V\right)=\mathrm{Concat}({\mathrm{head}}_{1},\ldots ,{\mathrm{head}}_{h})W^{o}$$
where ${\mathrm{head}}_{i}=\mathrm{Attention}\left(Q_{i},K_{i}{,V}_{i}\right)=\mathrm{Attention}\left({\mathrm{AW}}_{i}^{q},{\mathrm{AW}}_{i}^{k},{\mathrm{AW}}_{i}^{v}\right)$, $W^{o}\;\in \;R^{hd_{v}\times d_{a}}$ is a parameter matrix for the output, and h is the head number.

Although the self-attention mechanism offers the advantage of faster parallel computing for time-series data compared to the serial computing mode of LSTM, it does not consider the ordering information of each timestep in the time sequence due to the parallel computing method. Thus, positional encoding information is added upstream of the multi-head attention module to distinguish different input timesteps by assigning them unique values. The positional encoding method is implemented in the same manner as the equations used in the original Transformer paper [27]. The design of the feed-forward module is also identical; the first dense layer is followed by an ReLU activation function, and the second dense layer is not subsequently connected to a neuron activation function. Dropout layers are added to prevent the overfitting of the model, and the drop rate is set at 0.1. Residual connection reduces the difficulty of training a deep network by concatenating the layer input with the layer output, in case the learning outcomes in this layer are ineffective [34]. The global average polling layer reduces the dimensionality of the obtained time sequence to fit in the subsequent dense layers. After that, a branched model structure can be observed near the output layer for two joint angle predictions.

### 2.4. Model Training

Model training was conducted in the Python programming environment with the deep learning framework TensorFlow 2.7.0, and a single GPU (NVIDA GeForce RTX 3060) was equipped to accelerate the training process. The model size of the baseline and attention models are comparable (174,978 params versus 171,922 params) to ensure a similar level of computational complexity. The mean square error (MSE) is selected as the model cost function for training, and the model parameters are optimized using an Adaptive Moment Estimation (Adam) optimizer [35]. Learning rate decay is used to satisfy the need for a smaller update amount for the model parameters when approaching the solution point for minimum error. Early stopping is employed to terminate model training once there is no discernible error decrease, which helps prevent model overfitting. The selected metrics for evaluating the model fit include the normalized root mean square error (NRMSE), root mean square error (RMSE), and mean absolute error (MAE). Among the three metrics, NRMSE represents the model error in percentage terms, which may provide a more intuitive indication of the magnitude of the model error. Due to varying ranges of movement in limb joints, such as the ankle and knee, normalized RMSE allows the prediction errors across different joints to be compared. A smaller RMSE in a limb joint could be simply due to its limited range of motion.

NRMSE is defined as:(6)$$\mathrm{NRMSE}=\frac{\sqrt{\frac{1}{n}\sum_{j=1}^{n}{\left(y_{j}-{\hat{y}}_{j}\right)}^{2}}}{\max\left(y\right)-\min(y)}$$
where y is the ground truth of the entire dataset, $y_{j}$ is the ground truth of the jth data point, ${\hat{y}}_{j}$ is the prediction value of the jth data point, and n is the total number of data points.

RMSE is defined as:(7)$$\mathrm{RMSE}=\sqrt{\frac{1}{n}\sum_{j=1}^{n}{\left(y_{j}-{\hat{y}}_{j}\right)}^{2}}$$

MAE is defined as:(8)$$\mathrm{MAE}=\frac{1}{n}\sum_{j=1}^{n}\left|y_{j}-{\hat{y}}_{j}\right|$$

### 2.5. Transfer Learning

The lack of participant variety in the database impairs the deep learning model’s performance. Although the advanced model architecture is beneficial, many researchers are transitioning from a model-centric approach to a data-centric one [36]. Transfer learning is a strategy in the field of deep learning that, by sharing prior knowledge between relevant tasks, can greatly improve the generalizability of the pre-trained model [37]. Since the pre-trained model has learned to extract the fundamental features of motion patterns from previous subjects, only a small proportion of new subject data is required to fine tune and specialize the model [32]. In this research, we regard the previously well-trained baseline models and attention models as the pre-trained models. These pre-trained models have never seen the testing subject before, and they will continuously be refined using two schemes, namely 4% and 20% of the data from the testing subject, respectively. The two values were selected as the two reasonable extremes, i.e., only two trials or as high as ten trials. The refined models are then put to the test using the test subject’s remaining trials. The purpose of the section on transfer learning is to illustrate the importance of including the specific user’s data. In reality, a large walking data pool should be created for model training, which incorporates walking patterns that are similar to those of the user in question.

### 2.6. Paired T-Test

Several descriptive statistical variables, such as the mean error value, can be used to compare whether an apparent improvement in performance exists from the baseline model to the attention model. However, the utilization of statistical inference is more rigorous to make such a judgement. The paired *t*-test is a commonly used statistical test that compares the means of two related groups to determine if there is a statistically significant difference between them [38]. In pursuit of enhanced objectivity, this research mainly conducts this paired *t*-test to verify the effectiveness of the attention model in error reduction. Because a 10-fold testing scheme has been used, the baseline model and attention model each have ten testing subject results that can be paired into two groups. The Shapiro–Wilk test is used first to assess the normality of the difference between the two groups. If the difference between groups follows a normal distribution, a paired *t*-test will be used to determine whether the impact caused by the attention model is discernible. Otherwise, the Wilcoxon signed-rank test, a non-parametric statistical hypothesis method, should be performed [38,39]. Not only are the original baseline model and attention model compared, but also the fine-tuned baseline model and attention model, to verify if the beneficial effect of the attention model is diminished after applying transfer learning. All statistical analyses are implemented using the Python package SciPy 1.7.3.

### 2.7. Frequency Analysis

The noise (i.e., unexpected high frequency components) in the model predictions should be as low as possible for controllability and safety reasons. To understand whether the attention model exhibits smoother predictions compared to the baseline model, we employ FFT analysis to examine the frequency domain distribution of the model’s output. This analysis includes not only the original baseline model and attention model, but also all the fine-tuned models.

## 3. Results

### 3.1. Model Evaluation

With a 10-fold leave-one-out testing scheme, the baseline and attention models will each have ten testing results. Three error performance metrics (NRMSE, RMSE, and MAE) for evaluating ankle and knee joint prediction are adopted, and the corresponding mean value and standard deviation based on the ten testing results combined are presented in Table 1. These descriptive statistical results indicate that the attention model outperforms the baseline model in all evaluation metrics for either knee or ankle angle estimation. The lower NRMSE errors indicate that the knee angle is less difficult to predict than the ankle angle; NRMSE eliminates the effect of different movement ranges for the ankle and knee and so makes their prediction errors comparable. Illustrative model predictions are plotted in Figure 7 and Figure 8. Two different walking routes are considered: one with stair ascent (SA) and ramp descent (RD), and another with stair descent (SD) and ramp ascent (RA); both routes have sitting (S), standing (ST), and level walking (LW). The ability to handle activity transitions appears to be reasonable. It can also be seen that the prediction errors primarily occur at the peak points.

### 3.2. Fine-Tuned Models

Two transfer learning scenarios are investigated: fine-tuning the pre-trained models with either 4% or 20% of the testing subject’s data. After that, the fine-tuned models will be tested using the remaining testing trials. A comparison of the pre-trained and fine-tuned models can be made by comparing Table 1 and Table 2. With the application of transfer learning techniques, all three evaluation metrics show a promising reduction in the testing error for both the ankle and knee angles. The fine-tuned models also show smaller error standard deviations than the pre-trained models, both in the baseline and attention model architectures, indicating improved generalizability for different testing subjects. In comparison to training a new model from the beginning with the entire training set, the time required for fine tuning the pre-trained model may be disregarded because transfer learning only requires a small number of data samples and minimal computational load.

### 3.3. Statistical Tests for Model Performance

The testing errors of the baseline model and attention model, either original or finetuned, are depicted in Figure 9 to illustrate the prediction error distribution under a 10-fold leave-one-out scheme. Before conducting the paired *t*-test, the normality of the difference between the two sets of model errors has been tested using the Shapiro–Wilk test, and the relevant *p*-values are presented in Table 3. Almost all cases (*p*-value > 0.05) can be considered to be drawn from a normal distribution, so paired *t*-tests are applicable. However, the fine-tuned case (20%) for knee angle prediction is an exception, and so it should be tested using the Wilcoxon signed-rank test instead.

The null hypothesis for both the paired *t*-test and Wilcoxon signed-rank test is that the attention model has no effect on reducing the model error when compared to the baseline model. Accordingly, the alternative hypothesis will be accepted if the *p*-value is less than 0.05, indicating that the attention mechanism affects model error reduction. Paired *t*-tests are conducted on the three model evaluation metrics as shown in Table 4, whereby a smaller *p*-value indicates more confidence that the error reduction due to the attention model is significant. For most scenarios in Table 4, the attention model architecture is shown to be effective because of the *p*-values < 0.05. However, as the degree of transfer learning increases, the benefits of the attention model architecture are less evident due to the increased *p*-value. In particular, the *p*-value of 0.051 for the knee angle estimation signifies that the attention model has no apparent effect in reducing error when compared to its baseline counterpart after being fine tuned with 20% data.

### 3.4. Frequency Analysis

From the time domain comparisons in Figure 7 and Figure 8, the waveforms in the baseline model can be seen to exhibit more high-frequency fluctuations. An analysis of the frequencies was conducted using the Fast Fourier Transform and is illustrated in Figure 10. This represents the power distribution of the model predictions in the frequency domain for the original and fine-tuned models. Usually, the most significant signal components associated with human activity are detected at frequencies well below 6 Hz [40]. Both the baseline and attention models produce lower noise after being fine tuned, and more obvious noise reduction can be observed with more intervention from transfer learning (i.e., with the 20% case compared to the 4% case). Moreover, for both ankle and knee angle estimation, the original and fine-tuned attention models have less high-frequency power than their baseline counterparts. Note that the ground truth angles have been low-pass filtered by the dataset author [31] at 10 Hz, and so the frequency content above 10 Hz is expected to be small.

## 4. Discussion

In comparison to earlier publications, the accuracy of our models is promising. Rai et al. [25] showed a 4.4° error (RMSE) in the prediction of the right ankle angle for an unseen testing subject during walking activities on stairs and flat ground. However, numerous input features are used in their LSTM model to accomplish a small error, including the angular position of 21 full-body joints in three anatomical planes (sagittal, frontal, and transverse). Rai et al. [5] also evaluated the efficacy of an LSTM model on an obstacle avoidance dataset, and they estimated the ankle joint angle with an error of 5.5° (RMSE). However, their advanced attention model yields inferior results (9.1° RMSE), so that they added the history of the target joints as model inputs to reduce error. Sharma and Rombokas [39] used the joint angles of 19 joints in three anatomical planes and egocentric vision to train a LSTM model, and the relevant data were collected on stairs, in classrooms, and in atriums. Their overall prediction errors for ankle and knee angle are 11.9% (±5.2%) and 12.9% (±5.7%) for the NRMSE, respectively. Boudali et al. [1] used the wearer’s right upper limb together with a cane to estimate the contralateral knee angle (11.37° RMSE), although the testing environment was limited to flat ground, stair ascent, and standing. As in our study, the aforementioned researchers tested their models with participants whose gait data had not been used during the training process. As shown in Table 1, the joint angle prediction errors achieved in this work are promising given the inter-subject testing scenario involving diverse locomotion tasks. Aside from the model error, the joint angles produced by our proposed attention model are less oscillatory compared to the counterpart baseline model, thereby having the potential to reduce discomfort for users. The *t*-tests indicate that the proposed attention model has a statistically significant effect on reducing prediction error.

The generalizability of a joint angle estimation model depends upon the inclusiveness of the training dataset to a considerable extent. This entails gathering data from a wide range of subjects with varying physiological features and gait styles. Due to the high cost of data collection, the majority of research studies employ a limited number of subjects, typically ranging from five to twenty. As a result, the scarcity of data restricts the potential of a data-driven deep learning model, and in this situation, prediction errors for a new subject might be large. Rai et al. [25] demonstrated experimentally that increasing the number of subjects is more important than the expansion of the database size using limited subjects. The transfer learning results have also demonstrated that even a small number of samples from the new testing subject can effectively improve testing performance. Although the attention model architecture always outperforms the baseline model in terms of error for both original and fine-tuned cases, this advantage diminished as the degree of transfer learning increases (i.e., from the 4% case to the 20% case). In recent years, model training has tended to shift from a model-centric to a data-centric mode, with highly sophisticated model architectures being increasingly replaced by high-quality data [36]. Especially considering the diversity of users’ gait styles in the biomechanics estimation area, a larger pool of subjects would enable the model to learn a wider range of different walking patterns. In summary, it is unlikely that the model accuracy demonstrated in this paper is yet sufficient for the successful control of walking assistive devices.

The central focus of this study involves constructing a trajectory planner and comparing it with existing data. Hardware deployment concerns regarding real-time signal acquisition and actuator control have not been investigated. Dey and Schilling [12] have studied the feasibility of onboard controller implementation by deploying a deep learning model on a Raspberry Pi 4. Their objectives were to generate real-time foot angular position, despite the lack of sensors or actuators in their prototype. Inaccurate sensor placement poses a risk to the accuracy of the model, necessitating the development of a robust model that exhibits reduced sensitivity to imprecise sensor positioning. To generate training data, we recommend avoiding fixing the sensor at a very precise location on the limb. Instead, a broader placement area is preferred to enhance model robustness and diminish sensitivity to the specific location coordinates. In addition, undertaking feature engineering is valuable for assessing the significance of input features and eliminating superfluous sensor signals, thereby reducing the complexity of biomechanical signal collection.

## 5. Conclusions

This work explores the application of a deep learning model with a self-attention mechanism for the purpose of generating joint angle trajectories for use in controlling leg prostheses or orthoses. The intention is to create a more robust CLME as a flexible and self-decided motion, which is beneficial to improving human–machine interaction and adoption. This work also investigates several other Transformer-inspired deep learning techniques, such as positional encoding. Its application to a dataset encompassing 10 subjects and a wide range of walking modes demonstrates that the proposed attention model outperforms an LSTM baseline model in terms of both the prediction error and output smoothness. The proposed unified model shows the potential to continuously estimate the desired motion across various gait modes, avoiding control errors with assistive devices, which use traditional finite-state trajectory planning during sudden activity transitions, and the FFT results demonstrate that a more advanced attention mechanism is helpful to lower the noise level. The investigation of transfer learning demonstrates that using even a small amount of movement data from the unseen subject to refine the model does improve the model performance. This may indicate that a lack of data diversity in the original dataset restricts the model’s potential to be generalized. However, for users with lower limp impairment (including amputation) it is not possible to temporarily collect user data to fine tune the pre-trained model, which emphasizes the need for an extensive database that can accommodate all possible gait styles of users.

Several prospective research directions should be addressed in the future. The effect of the proposed approach on the user’s experience needs to be evaluated. As a first step, an emulated evaluation will be undertaken, both experimentally and using musculoskeletal model simulation tools, such as OpenSim. Then, lower-limb amputees can be recruited as participants to test a real-world application environment more accurately. Model deployment and low-level controllers are also worthy of more research, including issues such as computational load and online signal processing. Nevertheless, the deep learning model results presented here show potential for enabling the successful control of lower limb assistive devices in the future.

## Figures and Tables

**Figure 1 sensors-24-00211-f001:**
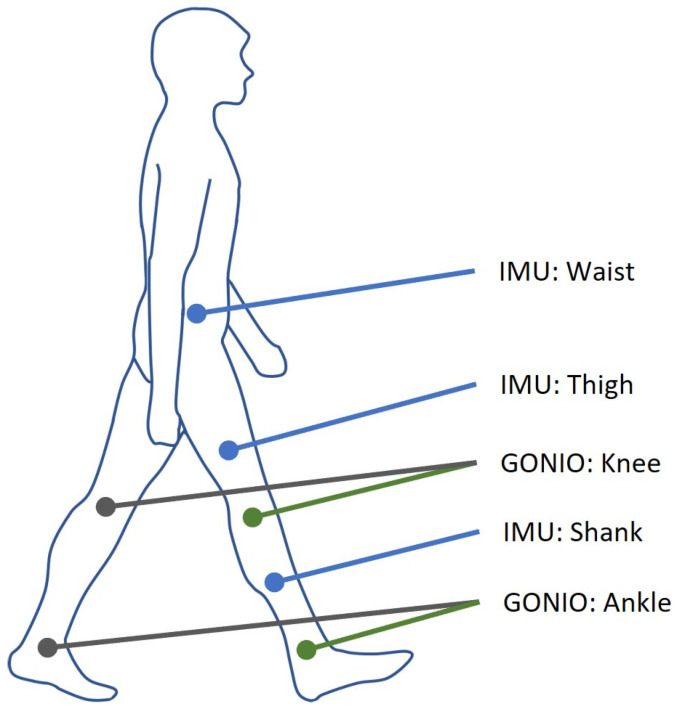
Sensor placement showing the data source from the subject body. The outputs to be predicted by the model are colored gray, while the model inputs are blue (IMU) and green (GONIO).

**Figure 2 sensors-24-00211-f002:**
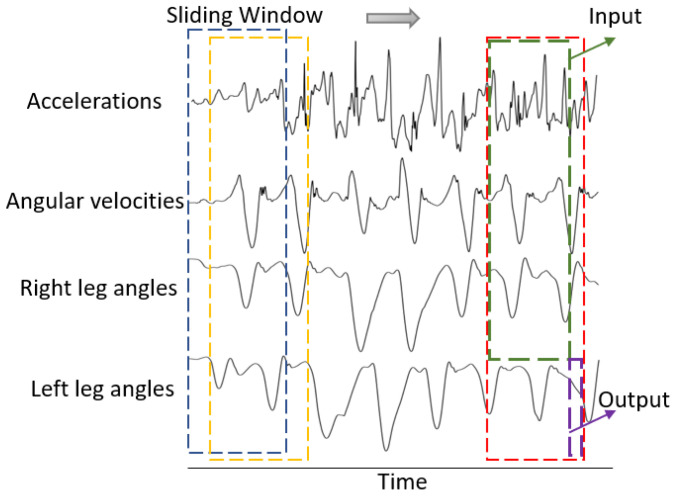
Sliding window to create time-series input data and targets. The sliding windows are represented by blue, yellow, and red rectangles. The accelerations and angular velocities measured by IMU sensors. The right ankle and knee angles are regarded as inputs (green rectangle); the left ankle and knee angles are used as targets (purple rectangle). Only four signals are plotted to illustrate the sampling process, due to limited space.

**Figure 3 sensors-24-00211-f003:**
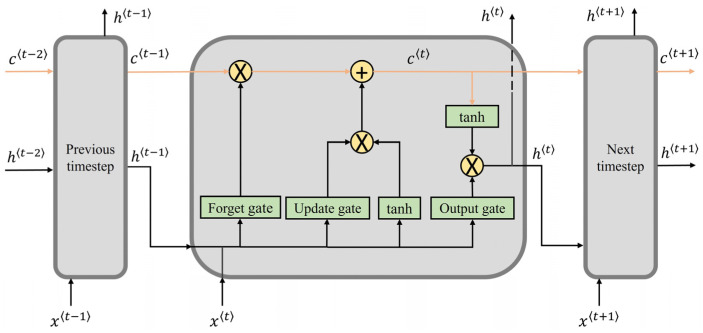
Internal operation of LSTM to process time-series data. Forget gate, update gate, and output gate are implemented using sigmoid function whose output is between 0 and 1, which represents the proportion of information retention. Specifically, $x^{\langle t\rangle }$ is the input of timestep t, $h^{\langle t\rangle }$ is the output of timestep t, and $c^{\langle t\rangle }$ is the cell state of timestep t.

**Figure 4 sensors-24-00211-f004:**
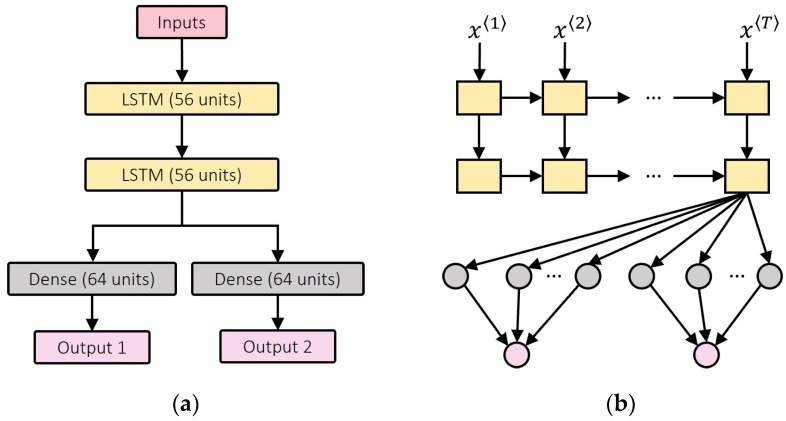
LSTM baseline model architecture. (**a**) Model layers and size. (**b**) Visualized information flow inside the model. In the Figure, the key components, including the LSTM layer, dense layer, and the information flow inside the LSTM layer, are represented for clarity. There are 25 timesteps in total (T = 25). The number of units is the number of neurons used in the neural network layer.

**Figure 5 sensors-24-00211-f005:**
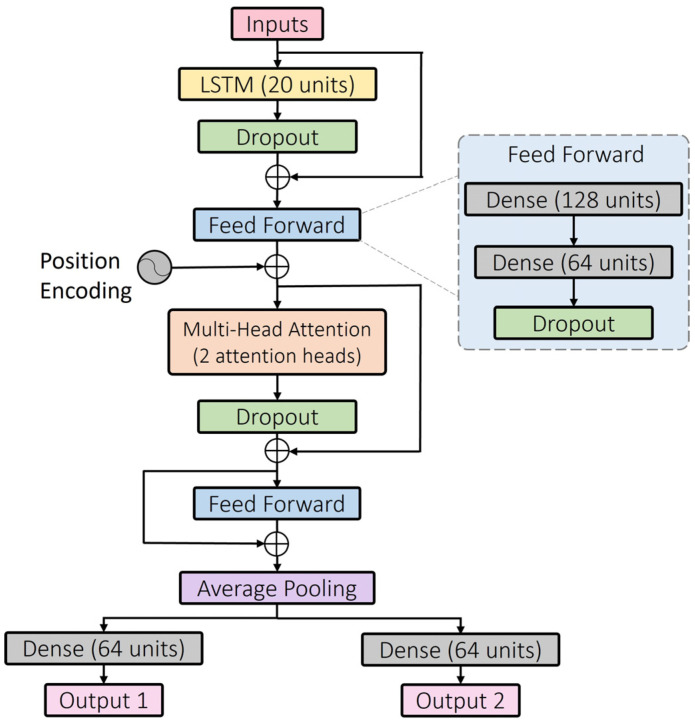
The proposed attention deep learning model. Several key components of the Transformer model [27] are adopted in the proposed model, including the multi-head self-attention mechanism, residual connection, positional encoding, and feed-forward module. The number of units is the number of neurons used in the neural network layer.

**Figure 6 sensors-24-00211-f006:**
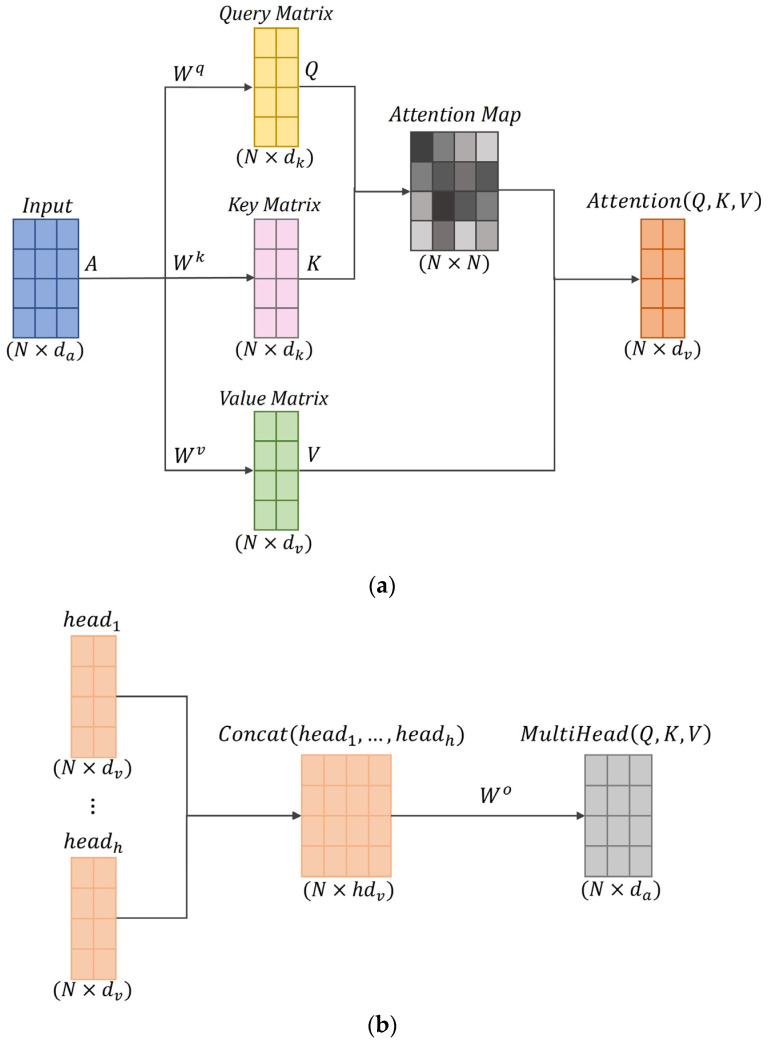
Calculation principle of the self-attention mechanism: (**a**) is about self-attention and (**b**) is for the multi-head self-attention. In the proposed attention layer, the number of input timesteps for the attention layer is N = 25, the number of heads is h = 2, and the dimensions are $d_{a}$ = 64, $d_{k}$ = 256, $d_{v}$ = 256.

**Figure 7 sensors-24-00211-f007:**
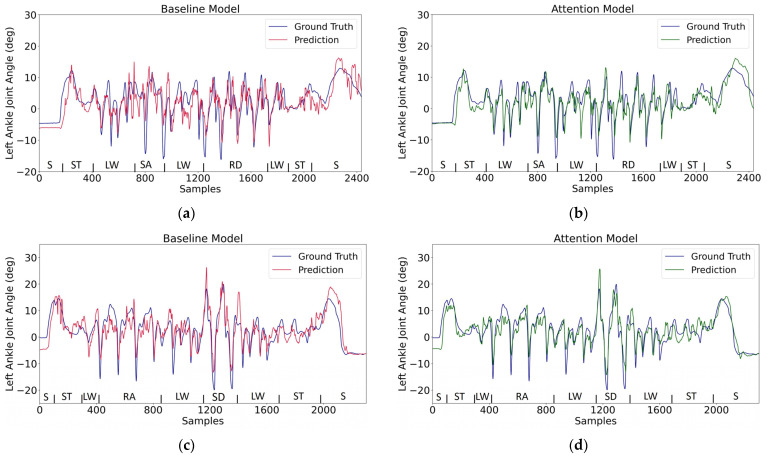
Model prediction for the left ankle angle on testing data across various activities (50 Hz sample rate). Two types of routes are tested: one route consists of sitting (S), standing (ST), level walking (LW), stair ascent (SA), and ramp descent (RD); another one consists of sitting (S), standing (ST), level walking (LW), ramp ascent (RA), and stair descent (SD). (**a**,**b**) Plot of the prediction for the first route; (**c**,**d**) plot of the prediction for the second route.

**Figure 8 sensors-24-00211-f008:**
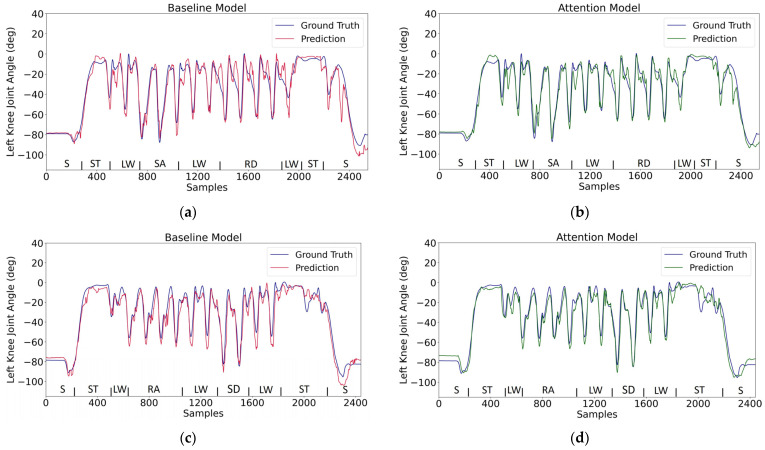
Model prediction for the left knee angle on testing data across various activities (50 Hz sample rate). Two types of routes are tested: (**a**,**b**) plot of the prediction for the first route; (**c**,**d**) plot of the prediction for the second route.

**Figure 9 sensors-24-00211-f009:**
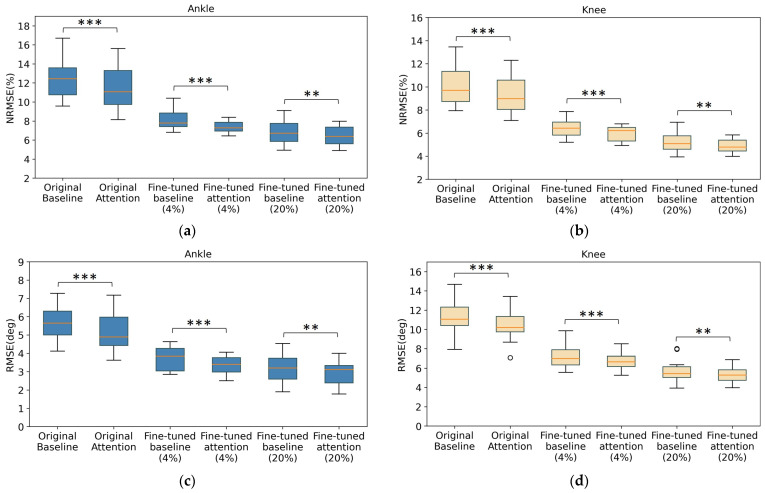
Box-whisker plots depicting the distribution of model errors. Three metrics are considered: (**a**,**b**) for NRMSE, (**c**,**d**) for RMSE, and (**e**,**f**) for MAE. Specifically, *** and ** indicate the difference between two cases is significant at the 1% and 5% levels, respectively.

**Figure 10 sensors-24-00211-f010:**
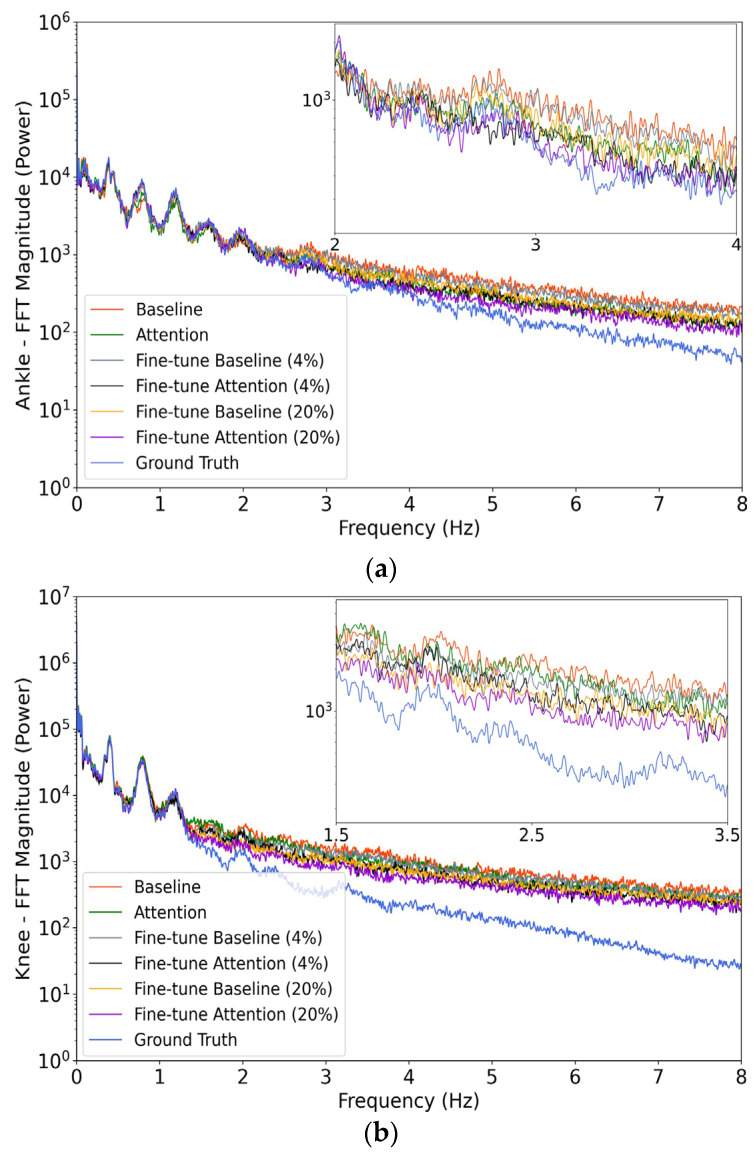
FFT analysis of the ground truth and model prediction. (**a**,**b**) Show the results for ankle and knee, respectively. The original and fine-tuned models are used.

**Table 1 sensors-24-00211-t001:** Model errors using the testing dataset.

Metric	Architecture	Ankle	Knee
NRMSE	Baseline	12.44% ± 2.20%	10.08% ± 1.80%
	Attention	11.50% ± 2.37%	9.31% ± 1.56%
RMSE (deg)	Baseline	5.63 ± 0.93	11.24 ± 2.05
	Attention	5.20 ± 1.07	10.38 ± 1.71
MAE (deg)	Baseline	4.59 ± 0.91	8.59 ± 1.74
	Attention	4.24 ± 1.04	7.86 ± 1.42

The data in the table are shown as mean value ± standard deviation.

**Table 2 sensors-24-00211-t002:** Fine-tuned model errors for the unseen testing dataset.

Metric	Architecture	Ankle	Knee
NRMSE	Fine-tuned Baseline (4%)	8.24% ± 1.15%	6.44% ± 0.85%
	Fine-tuned Attention (4%)	7.39% ± 0.64%	5.96% ± 0.67%
	Fine-tuned Baseline (20%)	6.88% ± 1.38%	5.21% ± 0.91%
	Fine-tuned Attention (20%)	6.45% ± 1.07%	4.88% ± 0.58%
RMSE (deg)	Fine-tuned Baseline (4%)	3.75 ± 0.67	7.19 ± 1.19
	Fine-tuned Attention (4%)	3.36 ± 0.47	6.65 ± 0.91
	Fine-tuned Baseline (20%)	3.13 ± 0.78	5.73 ± 1.28
	Fine-tuned Attention (20%)	2.94 ± 0.66	5.35 ± 0.89
MAE (deg)	Fine-tuned Baseline (4%)	2.86 ± 0.54	5.13 ± 0.76
	Fine-tuned Attention (4%)	2.55 ± 0.38	4.74 ± 0.67
	Fine-tuned Baseline (20%)	2.34 ± 0.63	3.97 ± 0.90
	Fine-tuned Attention (20%)	2.20 ± 0.52	3.77 ± 0.62

The data in this table are shown as mean value ± standard deviation.

**Table 3 sensors-24-00211-t003:** Normality evaluated by the *p*-value.

Metric	Paired Model Group	Ankle	Knee
NRMSE	Original baseline and attention	0.574	0.705
	Fine-tuned baseline and attention (4%)	0.949	0.436
	Fine-tuned baseline and attention (20%)	0.597	0.099
RMSE	Original baseline and attention	0.797	0.780
	Fine-tuned baseline and attention (4%)	0.700	0.261
	Fine-tuned baseline and attention (20%)	0.517	**0.046**
MAE	Original baseline and attention	0.859	0.515
	Fine-tuned baseline and attention (4%)	0.893	0.923
	Fine-tuned baseline and attention (20%)	0.570	0.548

The value in **bold** represents the case that cannot be accepted as normally distributed.

**Table 4 sensors-24-00211-t004:** Model performance difference evaluated by the *p*-value.

Metric	Paired Model Group	Ankle	Knee
NRMSE	Original baseline and attention	0.002	<0.001
	Fine-tuned baseline and attention (4%)	0.002	0.003
	Fine-tuned baseline and attention (20%)	0.023	0.015
RMSE	Original baseline and attention	<0.001	<0.001
	Fine-tuned baseline and attention (4%)	0.002	0.004
	Fine-tuned baseline and attention (20%)	0.021	0.019
MAE	Original baseline and attention	0.002	<0.001
	Fine-tuned baseline and attention (4%)	0.002	0.006
	Fine-tuned baseline and attention (20%)	0.029	**0.051**

The value in **bold** represents the case that the null hypothesis cannot be rejected (*p*-value > 0.05).

## Data Availability

The dataset analyzed in this study is openly available in FigShare at https://doi.org/10.6084/m9.figshare.5362627.v2 (accessed on 27 December 2023).

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
