# Peer review of "A Deep Learning Model with a Self-Attention Mechanism for Leg Joint Angle Estimation across Varied Locomotion Modes"

_sensors, 2023, doi:10.3390/s24010211_

Round 1

Reviewer 1 Report

Comments and Suggestions for Authors

I am looking forward to seeing your revisions.

Comments on the Quality of English Language

Extensive editing for clarity is required as the language is convoluted

Author Response

Thank you for your comments. Please see attached document for detailed replies.

Reviewer 2 Report

Comments and Suggestions for Authors

1. How does the self-attention mechanism in the temporal deep learning model improve the generation of lower limb joint angle trajectories compared to conventional finite state strategies?

2. What challenges are associated with using electromyography (EMG) for motion tracking, and how does the study address these issues? How does attaching IMU sensors to specific body parts (waist, thigh, shank) contribute to accurately measuring lower limb movement?

3. How does transfer learning enhance the generalizability of the model, and what was the rationale behind choosing the specific proportions (4% and 20%) of new subject data for fine-tuning?

4. Can you describe the paired t-test methodology used to compare the baseline and attention models, and how does this statistical method contribute to validating the effectiveness of the attention model?

5. How does the multi-head self-attention mechanism in the proposed model differ from traditional self-attention mechanisms, and what advantages does this modification bring?

6. Can you elaborate on how positional encoding is implemented in the multi-head attention module and why it's necessary for handling time series data?

Please read these papers to calculate the joint angles in human joints via different approaches.

H. Baek, A. M. Khan, V. Bijalwan, S. Jeon and Y. Kim, "Dexterous Robotic Hand Based on Rotational Shape Memory Alloy Actuator-Joints," in IEEE Transactions on Medical Robotics and Bionics, vol. 5, no. 4, pp. 1082-1092, Nov. 2023, doi: 10.1109/TMRB.2023.3315783. 

Comments on the Quality of English Language

The English quality of the manuscript is ok.

Author Response

Thank you for your comments. Please see the attached document for detailed replies.

Reviewer 3 Report

Comments and Suggestions for Authors

1. The authors should split the abstract into subsections to increase the soundness of the Research study.

2. The introduction section must contain the research problem, motivation and contribution of the proposed study.

3. The results must be compared with other existing research approaches. 

4. Explain the scope of the research study and the proposed method.

5. The  dataset encompassing 10 subjects and a wide range of walking modes demonstrates that the proposed attention  model outperforms an LSTM baseline model in terms of both prediction error and output  smoothness. what are the ten types of subjects covered by completing this study.

Comments on the Quality of English Language

The article must be properly checked. The typos and grammatical mistakes must be removed.

Author Response

(The authors gave the same response as above.)

Round 2

Reviewer 1 Report

Comments and Suggestions for Authors

I would like to thank you for your thoughtful responses to my requests.  Please receive my apologies for not being kinder in some of my comments.

1)     If this is a machine learning paper, as you have stressed in your response, then your description of the machine learning algorithms needs to be more in depth in the introduction.  The high level controller information is interesting, however, these previous controls do not reflect where the authors are going with this manuscript. Please consolidate the paragraphs between lines 48 and 87 into a single brief paragraph.

2)     Please consider moving the first paragraph of the methods into the introduction, as this fits better there, if you are going to keep the information about the drawbacks of EMG in the manuscript. 

Line 224. LSTM has already been defined.

Thank you again for your hard work in the revisions of this manuscript.  Best of Luck.

Author Response

Please see the attached file for details.

Regards,

Ioannis
